**Data Availability Statement:** Non-sensitive data underlying the results and code to reproduce the analysis presented in the study are available at

# Differential impact of the COVID-19 pandemic on laboratory reporting of norovirus and *Campylobacter* in England: A modelling approach

**Nikola Ondrikova**[1,2,3]*, **Helen E. Clough**[1,3], **Amy Douglas**[4], **Miren Iturriza-Gomara**[5], **Lesley Larkin**[4], **Roberto Vivancos**[3,6,7], **John P. Harris**[1,8], **Nigel A. Cunliffe**[1,3]

1 Institute of Infection, Veterinary and Ecological Sciences, University of Liverpool, Liverpool, United Kingdom, 2 Institute for Risk & Uncertainty, University of Liverpool, Liverpool, United Kingdom, 3 NIHR Health Protection Research Unit in Gastrointestinal Infections, University of Liverpool, Liverpool, United Kingdom, 4 Gastrointestinal Pathogens Unit, National Infection Service, Public Health England, London, United Kingdom, 5 Centre for Vaccine Innovation and Access, PATH, Geneva, Switzerland, 6 Field Service, National Infection Service, Public Health England, Liverpool, United Kingdom, 7 NIHR Health Protection Research Unit in Emerging and Zoonotic Infections, University of Liverpool, Liverpool, United Kingdom, 8 North West Health Protection Team, Public Health England, Liverpool, United Kingdom

* nikola.ondrikova@gmail.com

## Abstract

### Background

The COVID-19 pandemic has impacted surveillance activities for multiple pathogens. Since March 2020, there was a decline in the number of reports of norovirus and *Campylobacter* recorded by England's national laboratory surveillance system. The aim is to estimate and compare the impact of the COVID-19 pandemic on norovirus and *Campylobacter* surveillance data in England.

### Methods

We utilised two quasi-experimental approaches based on a generalised linear model for sequential count data. The first approach estimates overall impact and the second approach focuses on the impact of specific elements of the pandemic response (COVID-19 diagnostic testing and control measures). The following time series (27, 2015–43, 2020) were used: weekly laboratory-confirmed norovirus and *Campylobacter* reports, air temperature, conducted Sars-CoV-2 tests and Index of COVID-19 control measures stringency.

### Results

The period of Sars-CoV-2 emergence and subsequent sustained transmission was associated with persistent reductions in norovirus laboratory reports (p = 0.001), whereas the reductions were more pronounced during pandemic emergence and later recovered for *Campylobacter* (p = 0.075). The total estimated reduction was 47% - 79% for norovirus (12–43, 2020). The total reduction varied by time for *Campylobacter*, e.g. 19% - 33% in April, 1% - 7% in August.

https://doi.org/10.5281/zenodo.5035653. Raw norovirus and Campylobacter data were replaced with synthetic (model-generated) data. Routine surveillance data cannot be shared publicly because the provision of the data is dependent on the intended use. Raw norovirus and Campylobacter data are available from Public Health England (EEDD@phe.gov.uk).

**Funding:** NO is funded by EPSRC and ESRC Centre for Doctoral Training in Quantification and Management of Risk & Uncertainty in Complex Systems & Environments - Grant No. (EP/L015927/1).Funding website: https://gow.epsrc.ukri.org/NGBOViewGrant.aspx?GrantRef=EP/L015927/1 The funders had no role in study design, data collection and analysis, decision to publish, or preparation of the manuscript.

**Competing interests:** The authors have declared that no competing interests exist.

## Conclusion

Laboratory reporting of norovirus was more adversely impacted than *Campylobacter* by the COVID-19 pandemic. This may be partially explained by a comparatively stronger effect of behavioural interventions on norovirus transmission and a relatively greater reduction in norovirus testing capacity. Our study underlines the differential impact a pandemic may have on surveillance of gastrointestinal infectious diseases.

## Introduction

The impact of the COVID-19 pandemic has been felt at many levels beyond the direct consequences of illness and death from the Sars-CoV-2 virus. In England, laboratory reports of both norovirus and *Campylobacter spp.* are recorded via the national laboratory surveillance system (Second-Generation Surveillance System, SGSS); only *Campylobacter* is a notifiable causative agent under the Health Protection (notification) Regulations of 2010 [1]. In March 2020, a reduction was observed in the number of norovirus and *Campylobacter* laboratory reports to SGSS. The Emergency Department (ED) syndromic surveillance indicators reported by Public Health England also showed a decrease in ED attendances for all gastrointestinal illnesses during the same period [2].

This study focuses on norovirus and *Campylobacter* in England as these are the most common viral and bacterial causative agents of Infectious Intestinal Disease (IID), respectively [3]. Norovirus is responsible for the majority of gastroenteritis outbreaks in semi-enclosed settings such as hospitals [4] and care homes [5] in England, and the overall burden exceeds that of all other IID–causing pathogens [6]. The estimated annual economic cost of norovirus infections (£63 - £106 million) is higher than for *Campylobacter* (£33 - £75 million) [7]. Outbreaks of GI disease caused by *Campylobacter* infections are occasionally reported, and might be underestimated [8], but the majority of infections reported to national surveillance are defined as sporadic cases.

Campylobacteriosis is usually associated with consumption of undercooked food and cross-contamination during food preparation, particularly with handling chicken, but several other types of animal products have also been implicated in transmission [9]. Norovirus is predominantly associated with person-to-person transmission, although foodborne outbreaks due to contaminated food products (predominantly shellfish) or infected food handlers do occur [10]. Both pathogens display yearly seasonal effects; while norovirus activity is likely directly dependent on weather factors such as temperature [11], human *Campylobacter* infection depends on weather factors indirectly, through other mediating factors such as weather-related changes in human behaviour [12].

Previous studies have shown that the IID incidence derived through routine national surveillance underestimates the true disease burden [6]. Specifically, for every norovirus case detected by routine surveillance, another 288 cases (CI 239–346) are unreported in the community [6]. The corresponding figure is 1 in 9 (CI 6–14) for *Campylobacter* [6]. This study aims to assess the impact of the pandemic on the laboratory surveillance of norovirus and *Campylobacter* in England. We also explored the effect on surveillance of public health measures introduced during the pandemic; specifically, we investigated the relationship between laboratory reporting of norovirus and *Campylobacter* and (i) the number of Sars-CoV-2 tests conducted; and (ii) the stringency of infection prevention and control measures implemented at various points during the pandemic. To the best of our knowledge, this is the first study to

compare, both quantitative and qualitative differences in the pandemic's impact on the number of norovirus and *Campylobacter* reports. Additionally, the analysis provides insights into how to account for the pandemic in detection algorithms and predictive models used more broadly in public health.

## Materials and methods

### General approach and data utilised in the study

We utilised two quasi-experimental approaches to examine the reduction in laboratory reports of norovirus and *Campylobacter* in England that occurred since the emergence of Sars-Cov-2. The first approach aimed to estimate the overall decrease of laboratory reports and identify the type of the impact. The second approach examined the impact of the COVID-19 pandemic on laboratory reporting, utilising the stringency index to indicate the intensity of infection control and prevention measures and related changes in healthcare-seeking behaviour, and the number of Sars-Cov-2 tests conducted.

Weekly laboratory report totals for norovirus and *Campylobacter* between week 27, 2015 and week 43, 2020 were extracted from the national laboratory reporting surveillance system. Additionally, the Central England Temperature (CET) was used to indicate air temperature across England. The CET is a daily measure produced by the national meteorological service; we used weekly mean values to match the granularity of the laboratory reports.

Then, COVID-19 related data such as conducted Sars-CoV-2 tests and indicators of COVID-19 control measures stringency were considered. Specifically, data on testing for Sars-CoV-2 in England performed at diagnostic laboratories were used to indicate the pressure on testing services and the capacity to carry out regular activities. The Stringency Index is a measure to quantify the strictness of the government's response on a given day. It is based on nine control measures such as school and restaurant closures, stay at home orders and restriction on gatherings. The exact calculation is described elsewhere [13]. Tests were analysed as weekly sums and stringency index data as weekly means. Individual, explanatory time series together with data summaries and exploratory analysis are available on GitHub.

### Measurement of overall impact

The overall impact is estimated with simplified models which in one categorical indicator ($\delta$) represents the impact of policy decisions and change in human behaviour on reporting of both pathogens. The first COVID-19 death in the United Kingdom was reported in week 11 of 2020, national lockdown was announced in week 12 and started in week 13; hence three starting points (weeks 11, 12 and 13) are compared. Since our goal was to maximise the number of data points for the model to learn from, the considered period ends on week 43 of 2020.

This approach allows us to test whether the model, including the indicator, is significantly better than the model without it (H0). Two types of impact were tested [14]; i) level shift representing a consistent impact ($\delta = 1$), and ii) transient shift assuming exponential decay of the impact ($0 < \delta < 1$). For example, level shift is the same on the first as well as the fifth or tenth week, while for transient shift, the highest impact is assumed at the beginning, but decreases exponentially with time, e.g. ($\delta = 0.85$) in the second week but ($\delta = 0.52$) in the fifth week (see Fig 1). The estimated coefficient of the indicator is then a relative change in the weekly laboratory reports of a given pathogen, considering the other variables and effects in the model such as air temperature, seasonality and autoregression. To verify the significance of impact estimates, temporal falsification was performed. Figures of the expected trajectory for both pathogens, i.e. the expected number of laboratory reports had the pandemic not taken place, were based on the respective H0 models.

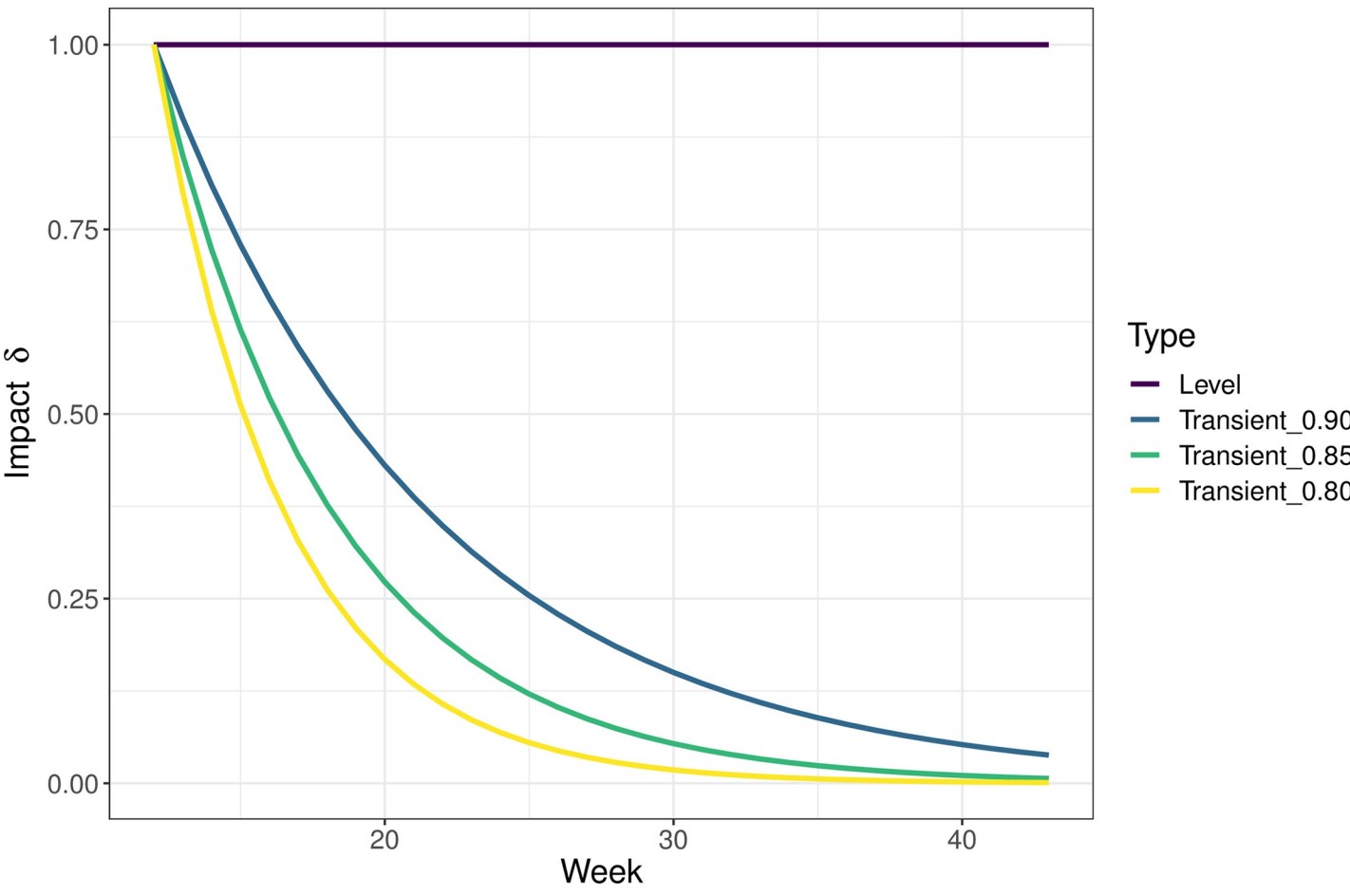

**Fig 1. Effect types considered in the comparison, W12-W43, 2020.**

### Measurement of specific trends

In order to identify specific trends, the COVID-19 pandemic was represented by using two variables: Sars-CoV-2 tests and stringency index. The residual impact was then captured by the impact indicator determined in the first (overall) model. Both of these variables were differenced to achieve stationarity. This was confirmed with the Ljung-Box test only for a shorter, 2-year period, and this period was therefore used in the sensitivity analysis. Specifically, a model was fitted to the shorter period and the point estimate was considered stable if it fell within the 95% confidence interval estimated from the longer period model.

It was assumed that the stringency of control measures would have a lagged effect, whereas testing would have impacted the diagnostic laboratories' capacity on the given week.

### Statistical analysis

All of the models were fitted with a GLM for count time series (TSGLM). The conditional distribution was chosen as negative binomial to account for over-dispersion, and the link function was logarithmic [15]. Furthermore, all the models consider autoregressive effects (number of reports in week $t$ depends directly on the number of reports in week $t-1$), two seasonal waves, a linear trend, indicators for Christmas and Easter Holidays and air temperature

**Table 1. Summary of the models.**

$log(\gamma_t) = intercept + linear\ trend + autoregression + [seasonal\ waves] + [explanatory\ variables]$

| | | Overall Impact | | Specific Trends | |
|---|---|---|---|---|---|
| **Explanatory variables** | **Description** | N | C | N | C |
| Air temperature $_{t-1}$ | Central England Temperature from previous week | ✓ | ✓ | ✓ | ✓ |
| Easter Holidays | Indicates weeks of Easter holidays | ✓ | ✓ | ✓ | ✓ |
| Christmas Holidays | Indicates weeks of Christmas holidays | ✓ | ✓ | ✓ | ✓ |
| Pandemic | Indicates weeks 11/12/13-43 of 2020, starting with the first death in the United Kingdom | ✓ | ✓ | ✓ | ✓ |
| Sars-CoV-2 testing | Number of tests at general diagnostic laboratories in a week (weeks 15–43, 2020; otherwise 0) | | | ✓ | ✓ |
| Stringency index $_{t-1}$ | Indicates lagged stringency of control measures against Covid-19 (weeks 3–42, 2020; otherwise 0) | | | ✓ | ✓ |

lagged by one week. This was determined based on the epidemiology and surveillance of both pathogens. A summary of all the models is provided in Table 1.

Models were assessed based on Akaike Information Criterion (AIC) and Logarithmic Score. Finally, all of the 95% level confidence intervals were obtained by parametric bootstrap. The analysis was performed using R [16] and the figures were produced with the R package 'ggplot2'[17]. The method used in this study is implemented in the 'tscount' R package, and it is described in detail in Liboschik, Fokianos and Fried [18]. The function used to estimate and to test the significance of the overall impact is explained here [14]. The code to reproduce the analysis is available on GitHub.

## Results

### Overall impact

The reduction in norovirus laboratory reports was significantly associated with the period after the first death from COVID-19 in the UK (week 11, 2020). The norovirus model assuming lagged effect of the first COVID-19 death, i.e. level shift starting in week 12, was better in terms of AIC (W11 = 2425.0, W12 = 2414.0, W13 = 2416.0) and logarithmic score (W11 = 4.32, W12 = 4.30, W13 = 4.30). The results for the *Campylobacter* models were similar, with level shift starting in weeks 11 and 12 being slightly better; AIC (W11 = 3452.3, W12 = 3452.7, W13 = 3458.3), logarithmic score (W11 = 6.17, W12 = 6.17, W13 = 6.18). Models assuming transient shift (i.e. exponential decay) starting in week 12 also showed better fit. For simplicity, only models assuming the start of the pandemic in week 12 will be discussed further.

Both pathogens showed a decrease in expected laboratory reports, but these effects were qualitatively different (Table 2). The reduction in the number of norovirus reports was best described by level shift (~59%; CI 51% - 67%), i.e. the impact of the pandemic was consistent over time (p = 0.001). The decrease in *Campylobacter* reports was better described by transient shift (δ = .85). The estimated impact was ~46% (CI 38% - 55%) on week 12, ~39% (32% - 47%) on week 13 and so on; the mean weekly reduction across weeks 12 and 43 was ~9% (CI 8% - 11%). This decrease was statistically significant at the 10% but not the 5% level (p = 0.075). As illustrated by Fig 2, this is likely because the effect was too short to be detected with a higher level of significance. The impact on norovirus was more pronounced.

To verify these results, temporal falsification was performed. In particular, the period between weeks 12 and 43, 2019 was used to estimate and test the significance of a best-fitting effect type, i.e. level shift for norovirus, transient shift for *Campylobacter* model. This period was not significantly associated with changes in the number of laboratory reports of norovirus or *Campylobacter*.

**Table 2. Overall impact of the COVID-19 pandemic (W12-W43, 2020) on norovirus and *Campylobacter*: Comparison of effect types, data from 2015–2020.**

| Effect Type | Norovirus | | | | Campylobacter | | | |
|---|---|---|---|---|---|---|---|---|
| | AIC | LogS | p | Reduction (CI) % | AIC | LogS | p | Reduction (CI) % |
| Level Shift (δ = 1) | 2414.0 | 4.30 | 0.001 | 59 (51–67) | 3452.7 | 6.17 | 0.192 | 12 (7–17) |
| Transient shift (δ = .90) | 2471.4 | 4.40 | 0.024 | 18 (15–21) | 3417.9 | 6.10 | 0.071 | 12 (9–15) |
| Transient shift (δ = .85) | 2492.2 | 4.44 | 0.045 | 9 (6–11) | 3417.5 | 6.10 | 0.075 | 9 (8–11) |
| Transient shift (δ = .80) | 2501.8 | 4.46 | 0.065 | 10 (7–11) | 3419.8 | 6.11 | 0.084 | 7 (6–8) |

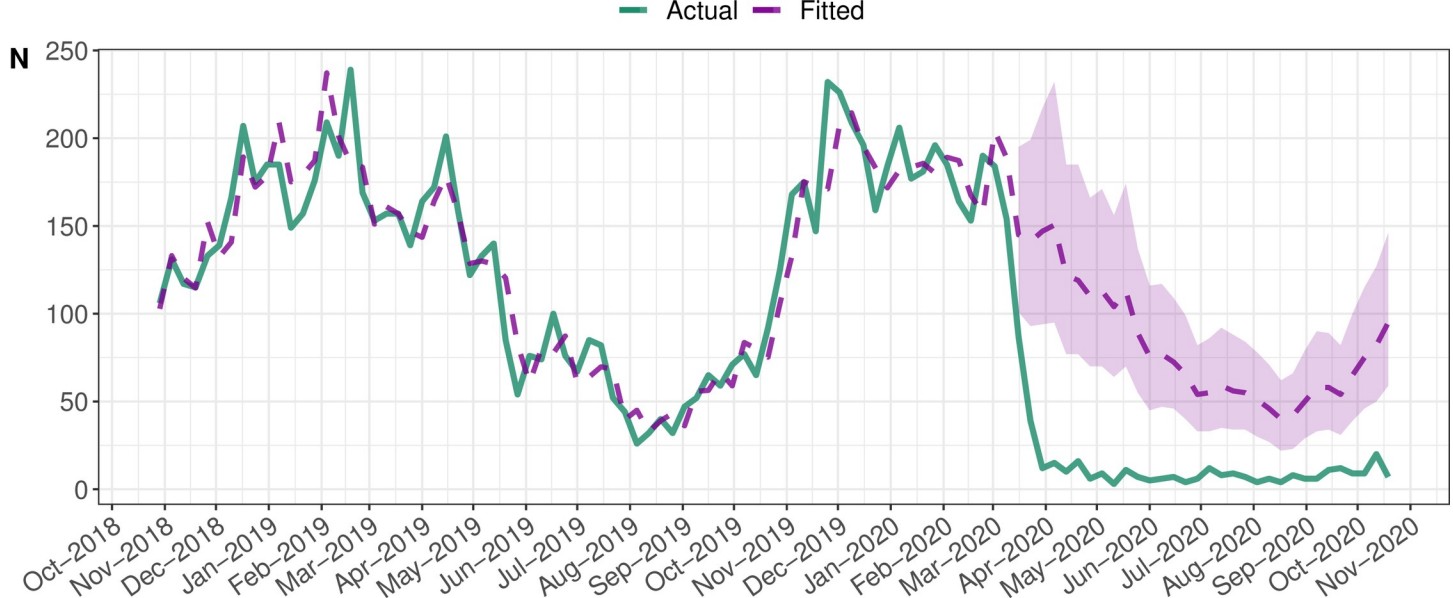

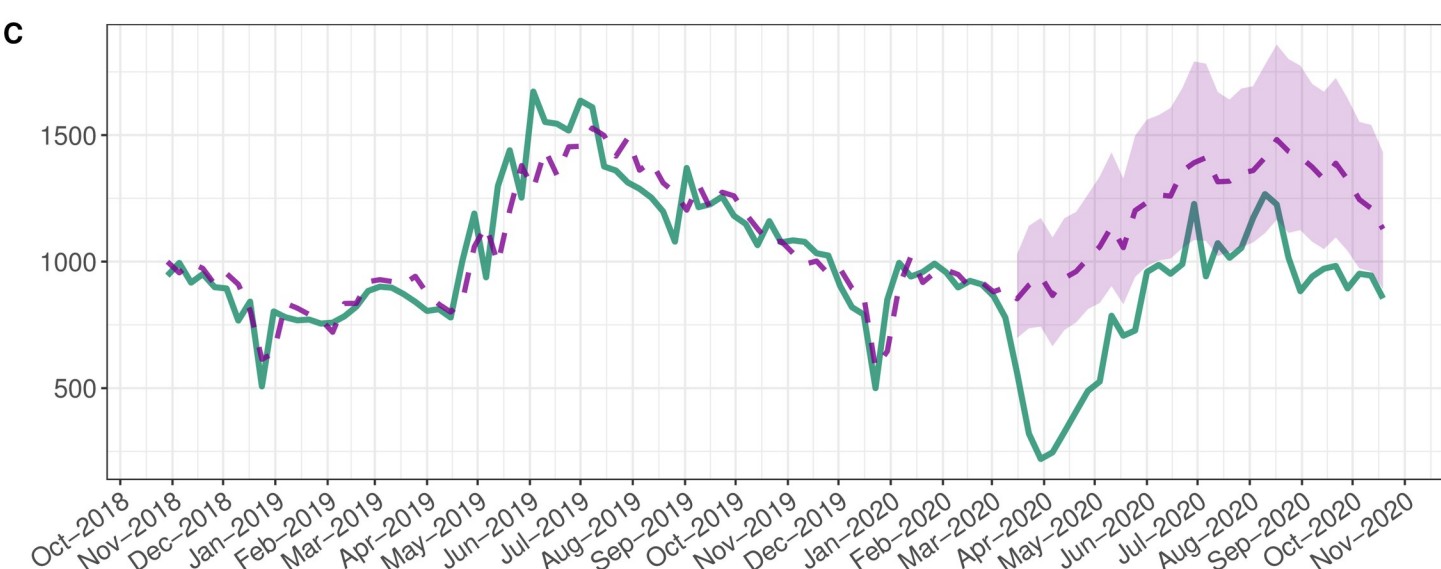

**Fig 2. Estimated (overall) impact of COVID-19 pandemic: Norovirus, *Campylobacter*, W12 –W43, 2020.** The figure displays the model's prediction (Fitted) and the actual number of weekly reports (Actual). The section with the ribbon, i.e. confidence interval, highlights what was expected in the absence of a pandemic.

**Table 3. Specific trends of the COVID-19 pandemic's impact (W11-W43, 2020) on norovirus and *Campylobacter*: Modelling coefficients, data from 2015–2020.**

| | Norovirus | | | Campylobacter | | |
|---|---|---|---|---|---|---|
| | *Estimate* | *CI(lower)* | *CI(upper)* | *Estimate* | *CI(lower)* | *CI(upper)* |
| Intercept | 2.226 | 1.887 | 2.683 | 2.564 | 2.192 | 3.11 |
| Ar (1) | 0.627 | 0.552 | 0.684 | 0.605 | 0.518 | 0.661 |
| Sin (2) | 0.015 | -0.028 | 0.059 | 0.011 | -0.01 | 0.031 |
| Cos (2) | -0.018 | -0.058 | 0.021 | 0.021 | 0.002 | 0.043 |
| Sin (4) | 0.019 | -0.02 | 0.063 | -0.001 | -0.02 | 0.021 |
| Cos (4) | 0.027 | -0.014 | 0.067 | -0.004 | -0.023 | 0.018 |
| Christmas | -0.078 | -0.266 | 0.1 | -0.362 | -0.455 | -0.26 |
| Easter | 0.048 | -0.101 | 0.196 | -0.083 | -0.165 | -0.006 |
| Linear trend | -0.004 | -0.019 | 0.026 | 0.005 | -0.005 | 0.015 |
| Air temperature (˚C) | -0.048 | -0.059 | -0.039 | 0.016 | 0.013 | 0.02 |
| **Sars-cov-2 tests** | **-0.002** | **-0.008** | **0.004** | **-0.001** | **-0.003** | **0.001** |
| **Stringency** | **-0.013** | **-0.031** | **-0.001** | **-0.006** | **-0.012** | **-0.001** |
| **Shift** | **-0.792** | **-1.015** | **-0.627** | **-0.551** | **-0.745** | **-0.399** |
| Overdispersion parameter | 0.035 | 0.027 | 0.046 | 0.012 | 0.01 | 0.015 |

"AR(1)" is an autoregressive term of order 1; "Sin(2)" and "Cos(2)" sinusoidal components to represent annual peaks; "Sin(4)" and "Cos(4)" similarly represent bi-annual peaks; "Christmas" and "Easter" are binary variables reflecting each period respectively; "Sars-cov-2" tests is the number of Sars-cov-2 tests conducted; Stringency stands for stringency index as defined in [13]; and "shift" is a level shift for norovirus, transient shift fo *Campylobacter* as determined by the Overall Impact model.

## Specific trends

The second modelling approach also demonstrated that norovirus was impacted relatively more during the early months of COVID-19 pandemic. Specifically, the relative effect of the stringency of the COVID-19 control measures was greater for norovirus laboratory reporting than for *Campylobacter*. In particular, changes in the stringency index were associated with a reduction of ~2% (CI 0% - 5%) on average (12–43, 2020) for norovirus and ~1% (CI 0% - 2%) for *Campylobacter*. Additionally, changes in testing capacity appear to have more negatively impacted norovirus reporting. As suggested by the model coefficient estimates (Table 3), norovirus laboratory reports decreased by ~2% (CI 0% - 10%) due to Sars-CoV-2 diagnostic testing on average every week while *Campylobacter* laboratory reports decreased by ~1% (CI 0% - 4%).

## Comparison: Overall impact, specific trends

The models, including specific trends, i.e. number of Sars-CoV-2 tests conducted and stringency index, on top of the shift variable, performed better in terms of the logarithmic score for weekly reports of norovirus (overall impact = 4.30, specific trends = 4.28) and *Campylobacter* (overall impact = 6.10, specific trends = 6.08). Similarly, in terms of AIC—2414.0 vs. 2412.0 for norovirus; 3417.5 vs. 3411.3 for *Campylobacter*.

A simpler (overall) model might have underestimated the impact as the total decrease in norovirus reports determined by the second (specific trends) model has wider confidence intervals ~59% (CI 47% - 79%). On the other hand, the point estimates of both the Overall Impact and Specific Trends models were ~59%. Similarly, the total mean reduction of *Campylobacter* reports estimated by the Specific Trends model was between ~11% (CI 8% - 17%), which is higher than the simpler Overall Impact model ~9% (8% - 11%). Note that the impact estimate from the Specific Trends models is a sum of all the pandemic related estimates, i.e.

conducted Sars-CoV-2 tests, stringency index and the shift determined by the overall impact model.

As both models including the specific trends showed better (i.e. lower) logarithmic score and AIC, these estimates will be discussed in the Discussion. The total estimated reduction was 47% - 79% for norovirus (12–43, 2020). The total reduction has changed in time for *Campylobacter*, e.g. 19% - 33% in April, 1% - 7% in August.

## Discussion

Our findings demonstrate that the reduction in laboratory reports of norovirus was significantly associated with changes in infection control policies and Sars-CoV-2 virus testing approaches consequent upon the emergence of the COVID-19 pandemic. The impact of the pandemic was more pronounced for weekly laboratory reporting of norovirus than laboratory reporting of *Campylobacter*. These impacts were qualitatively different; while *Campylobacter* reports noticeably decreased within the first weeks of the pandemic and later recovered (e.g. 19% - 33% in April, 1% - 7% in August), norovirus reports also decreased but then remained low (47% - 79%). Additionally, we found a stronger association of norovirus reports with changes in the stringency of COVID-19 control measures and the number of Sars-CoV-2 tests conducted, compared with *Campylobacter*.

The differential reduction in the reporting of norovirus and *Campylobacter* is likely explained by several reasons. Firstly, laboratory testing for norovirus was likely more impacted during the pandemic than was *Campylobacter*. The Royal College of Pathologists issued guidance [19] on halting the testing of non-bloody diarrhoea specimens, with which norovirus is typically associated. Additionally, the capacity to obtain samples for laboratory confirmation during IID-related outbreaks which are more commonly associated with norovirus [20], could potentially have been compromised by the pandemic. Overall, diagnostic laboratories likely prioritised Sars-CoV-2 testing over routine testing; of note, with increasing Sars-CoV-2 tests, norovirus laboratory reports decreased more compared with *Campylobacter*. A reduction in testing for norovirus and *Campylobacter* as well as other gastrointestinal pathogens was also reported in the USA [21].

Secondly, behavioural changes are likely to have impacted norovirus transmission more than *Campylobacter*. Norovirus infections are mostly transmitted person-to-person [22], and cause outbreaks in health and social care settings, with the greatest burden in care homes [5, 23], similar to the new coronavirus [24, 25]. On the other hand, risk factors for *Campylobacter* infection are mostly associated with foodborne transmission routes and poor food hygiene and handling, particularly with the consumption of under-cooked chicken [9, 26]. Considering these similarities between norovirus and Sars-CoV-2, there is likely to have been a true reduction in the incidence of norovirus resulting from infection control measures introduced for COVID-19 such as greater handwashing, social distancing and enhanced hygiene in care homes and other community and health care setttings. Regarding *Campylobacter*, restaurant closures due to the pandemic could have potentially reduced the transmission of infection, although food delivery was still available; an increase in preparation of food in the home, with the risk of inappropriate hygiene and under-cooking, could have had a more pronounced effect on increasing the risk of campylobacteriosis [26].

A further consideration is change in healthcare-seeking behaviour during the pandemic. Although laboratory reports of both pathogens decreased when the control measures against COVID-19 were more restrictive, this pattern was stronger for norovirus. A possible explanation is that norovirus and *Campylobacter* differ in clinical severity and duration of illness. Norovirus generally causes mild symptoms lasting 1–2 days [27]. Campylobacteriosis typically

lasts longer (1–5 days) and is associated with symptoms such as severe abdominal pain and bloody diarrhoea [28], meaning that patients with *Campylobacter* infection may be relatively more likely to contact healthcare services and to have a sample taken for laboratory diagnosis during the period in which pathogen confirmation was possible.

## Strengths and limitations

This study investigated the impact of the COVID-19 pandemic on the routine laboratory reporting of norovirus and *Campylobacter* using a quasi-experimental modelling approach; consideration of seasonality, autoregression and other factors helped to quantify the level of uncertainty. We were able to estimate the magnitude and direction of overall and specific impacts in terms of testing capacity and of behavioural changes via stringency of COVID-19 control measures, and were able to demonstrate that the impact of the pandemic differed qualitatively between norovirus and *Campylobacter*. We also showed that simply including a categorical indicator to capture the effect of the pandemic in existing models and algorithms might underestimate the impact and that additional variables such as the stringency index can be helpful.

This study has some limitations. Firstly, the analysis was performed on aggregated national data, and regional differences were not investigated. However, reliable estimation of regional level impact would be challenging, especially for norovirus due to the low numbers of laboratory reports in some regions and increasing uncertainty around the estimate. Modelling both pathogens brings many challenges; for example, specific risk factors and seasonality for *Campylobacter* can vary with age and different *Campylobacter* spp. [29], while for norovirus, season, age and certain annual events, such as the return to school after the summer break are considered to impact substantially on norovirus reporting. Norovirus is also more affected by underreporting and underascertainment [6], bringing additional uncertainty. Considering all of these challenges, our estimates might be conservative. Furthermore, we could not account for all the trends which might have affected the model estimates. For example, the increasing number of Sars-CoV-2 tests performed at the diagnostic laboratories at the period of time when there was lower testing capacity than later in the year coincides with the end of the norovirus season. Also, variables used in our analysis, such as stringency index, are proxies for what we hoped to estimate. In particular, we could not estimate the proportion of the impact attributed to specific factors such as a genuine reduction in transmission, changes in healthcare-seeking behaviour, etc.

## Conclusion

The number of reports of norovirus and *Campylobacter* fell significantly with the emergence of Sars-CoV-2. However, while laboratory reports of *Campylobacter* recovered, reports of norovirus remained low. The reasons are likely multifactorial, including differences in the transmission routes of these two pathogens. Since the predominant transmission route for norovirus is person to person, measures such as enhanced hand hygiene and enhanced infection prevention and control measures in social and healthcare settings, if maintained at a population level, could result in a sustained reduction in norovirus cases. Our study underlines the differential impact a pandemic may have on surveillance of gastrointestinal infectious diseases and so highlights that society's best efforts to control the pandemic infectious agent can have impacts above and beyond those that might be most immediately expected. This adds to the need for pandemic preparedness to include consideration of the maintenance of priority routine surveillance systems and the resource to analyse surveillance data during the pandemic period. The direct as well as indirect effects of the pandemic could, through impairing essential

surveillance functions, impede the ability to detect ongoing threats to national or international public health [30].

## Acknowledgments

Helen E. Clough, Roberto Vivancos, Nigel A. Cunliffe and Nikola Ondrikova are affiliated to the National Institute for Health Research (NIHR) Health Protection Research Unit in Gastro-intestinal Infections at University of Liverpool, in partnership with Public Health England, in collaboration with University of Warwick. The views expressed are those of the author(s) and not necessarily those of the NIHR, the Department of Health and Social Care or Public Health England.

We are very grateful to the editor and two reviewers (anonymous, Christian Bottomley) for their comments and suggestions.

## Author Contributions

**Conceptualization:** Nikola Ondrikova, Helen E. Clough, Amy Douglas, Miren Iturriza-Gomara, John P. Harris, Nigel A. Cunliffe.

**Data curation:** Amy Douglas, Lesley Larkin.

**Formal analysis:** Nikola Ondrikova.

**Funding acquisition:** Helen E. Clough, Miren Iturriza-Gomara, Roberto Vivancos, John P. Harris.

**Methodology:** Nikola Ondrikova.

**Supervision:** Helen E. Clough, Miren Iturriza-Gomara, Roberto Vivancos, John P. Harris, Nigel A. Cunliffe.

**Validation:** Nikola Ondrikova.

**Visualization:** Nikola Ondrikova.

**Writing – original draft:** Nikola Ondrikova.

**Writing – review & editing:** Helen E. Clough, Amy Douglas, Miren Iturriza-Gomara, Lesley Larkin, Roberto Vivancos, John P. Harris, Nigel A. Cunliffe.

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
