## [Decision Letter · Decision Letter 0]

25 May 2021

PONE-D-21-11454

Differential Impact of the COVID-19 Pandemic on Laboratory Reporting of Norovirus and Campylobacter in England: Modelling Approach

PLOS ONE

Dear Dr. Ondrikova,

Thank you for submitting your manuscript to PLOS ONE. After careful consideration, we feel that it has merit but does not fully meet PLOS ONE’s publication criteria as it currently stands. Therefore, we invite you to submit a revised version of the manuscript that addresses the points raised during the review process.

We look forward to receiving your revised manuscript.

Kind regards,

Shinya Tsuzuki, MD, MSc

Academic Editor

PLOS ONE

Journal Requirements:

2.We note that you have indicated that data from this study are available upon request. PLOS only allows data to be available upon request if there are legal or ethical restrictions on sharing data publicly. For information on unacceptable data access restrictions, please see http://journals.plos.org/plosone/s/data-availability#loc-unacceptable-data-access-restrictions.

3.We note that the grant information you provided in the ‘Funding Information’ and ‘Financial Disclosure’ sections do not match.

Additional Editor Comments:

One of the reviewers raised some concerns about modelling and I agree with his opinion. Please address the comments from both reviewers before further consideration.

Reviewers' comments:

Reviewer's Responses to Questions

**Comments to the Author**

1. Is the manuscript technically sound, and do the data support the conclusions?

Reviewer #1: Partly

Reviewer #2: No

2. Has the statistical analysis been performed appropriately and rigorously? 

Reviewer #1: No

Reviewer #2: No

3. Have the authors made all data underlying the findings in their manuscript fully available?

Reviewer #1: No

Reviewer #2: Yes

4. Is the manuscript presented in an intelligible fashion and written in standard English?

Reviewer #1: Yes

Reviewer #2: Yes

5. Review Comments to the Author

Reviewer #1: The study “Differential Impact of the COVID-19 Pandemic on Laboratory Reporting of Norovirus and Campylobacter in England: Modelling Approach” is interesting. The data and methodology parts are well described. I would recommend the authors add some more explanations and also describe the contribution of this work in the introduction section. Moreover, attention should be given to the following highlighted points before resubmitting.

1. Page 10 of 21, “The conditional distribution was chosen as negative binomial to account for over-dispersion, and the link function was logarithmic.” How the conditional distribution is selected any reference or source please provide details.

2. Page 13 of 21, “In particular, changes in the stringency index were associated with a reduction of ~7% (CI 2% - 6%)” check these confidence limits.

3. Table 3, the values of Sars-cov-2 tests is not correct for norovirus.

4. Construct a table that lists the character shorthand.

5. From Figure 1 it seems that the Actual data is following downward trend while the predicted data trend is upward. Is this a real picture of the data or correct modelling?

Reviewer #2: The authors present interesting surveillance data on the impact of lockdown restrictions on laboratory reports of norovirus and Campylobacter. The paper is very well written and broadly speaking the modelling approach seems reasonable. My main difficulty with the paper was that I found the results of the modelling hard to interpret in light of the data presented in Figure 2.

In the figure, norovirus cases reduce from ~200 per week pre-COVID to <20 cases per week post COVID, which is a >90% reduction. Why are the estimates presented in Table 2 significantly lower, even for delta=1?

Similarly, for Campylobacter the decrease is from ~1,000 to 250 which corresponds to a 75% reduction. But again the estimates are lower.

I agree with the authors assessment that delta=1 for norovirus and approximately 0.85 for Campylobacter but they do not provide any justification for the later value. You could argue, for example, that the reduction in Campylobacter halves over 4 weeks and 0.85^4=0.52.

It is a bit of shame that delta is not estimated from the data. Fitting a transfer function model would be one way to do this - see for example chapter 13 of “Time Series Analysis” by Box, Jenkins , Reinsel and Ljung (a pdf version of the book is available for free online)

In summary, I suspect the modelling needs to be revisited or better explained. If this can be addressed, then I think this will be an excellent paper.

6. PLOS authors have the option to publish the peer review history of their article (what does this mean?). If published, this will include your full peer review and any attached files.

Reviewer #1: No

Reviewer #2: **Yes: **Christian Bottomley

---

## [Author Response · Author response to Decision Letter 0]

9 Jul 2021

We appreciate the time and effort that the reviewers have dedicated to providing their valuable feedback. We have been able to incorporate changes to the analysis and the manuscript to reflect the comments provided. 

Point-by-point response to the reviewers’ comments and concerns. 

Comments from Reviewer 1 

• Comment 1: I would recommend the authors add some more explanations and also describe the contribution of this work in the introduction section 

Response: Even though, the contribution of the study is described in the discussion, I agree that it is good to mention it more in the introduction too. The highlighted text was added to the introduction. 

Changes in the manuscript (Introduction, lines 123-127):

… To the best of our knowledge, this is the first study to compare, both quantitative and qualitative differences in the pandemic’s impact on the number of norovirus and Campylobacter reports. Additionally, the analysis provides insights into how to account for the pandemic in detection algorithms and predictive models used more broadly in public health.

• Comment 2 (Page 10 of 21): “The conditional distribution was chosen as negative binomial to account for over-dispersion, and the link function was logarithmic.” How the conditional distribution is selected any reference or source please provide details.

Response: A negative binomial model is a standard approach to dealing with over-dispersion: for background context and to guide the reader we have inserted reference 15 was added (Materials and methods, line 193).

• Comment 3 (Page 13 of 21): “In particular, changes in the stringency index were associated with a reduction of ~7% (CI 2% - 6%)” check these confidence limits.

Response: Thank you for pointing this out – yes, this was indeed a typo. These values have further changed since the second model was updated in response to Referee 2 (Results, lines 249-254).

• Comment 4 (Table 3): The values of Sars-cov-2 tests is not correct for norovirus.

Response: The model was updated so this table has now changed.

• Comment 5 (Table 3): Construct a table that lists the character shorthand. 

Response: We have modified the caption to Table 3 to reflect the shorthand, since this seems to be the most parsimonious way to represent this information.

• Comment 6 (Figure 1): From Figure 1 it seems that the Actual data is following downward trend while the predicted data trend is upward. Is this a real picture of the data or correct modelling?

Response: The Figure 1 (now Fig. 2) shows what would have happened had there been no pandemic. While norovirus in England tends to have winter seasonality and is expected to decrease in spring, Campylobacter reports tend to increase in spring. However, due to the pandemic, this was disrupted and so it is expected that the prediction (what would have happened) and the actual data (what happened) will look different. 

Changes in the manuscript (Results section, line 228):

As illustrated by Fig 2, this is likely because the effect was too short to be detected with higher level of significance. 

Notes under Fig 2 (lines 239-242): The figure displays the model’s prediction (Fitted) and the actual number of weekly reports (Actual). The section with the ribbon, i.e. confidence interval, highlights what was expected in the absence of a pandemic.

Comments from Reviewer 2

• Comment 1 (Figure 2): In the figure, norovirus cases reduce from ~200 per week pre-COVID to <20 cases per week post COVID, which is a >90% reduction. Why are the estimates presented in Table 2 significantly lower, even for delta=1?

Response: We agree this needs clarification. The pandemic arrived in the UK at the time of the year when norovirus usually decreases. Since our modelling accounts for the seasonality, some decrease is expected and shouldn’t necessarily be attributed to the pandemic. Similarly, this applies to the increasing air temperature (weaker for Campylobacter) and effect of easter holidays (dip in reporting). Also, the model considers the autoregressive nature of the weekly reporting. Therefore, the observed difference between the lines does not equal the estimated impact of the pandemic. On top of that, the actual difference between the Actual and Fitted lines in the pandemic period is ~86% and since the model’s Mean Absolute Error (MAE) on non-pandemic period is ~20 reports the estimates we provide make sense. Even the model assuming the start of the pandemic on week 13 showed point estimate of 62% (CI 53% - 70%); confirming the stability of the estimate. 

Changes to the manuscript (Results, lines 265 - 280): 

To make the models easier to interpret and compare, an impact variable was added to the second model to capture residual effects of the pandemic unexplained by the proxies, i.e. number of Sars-CoV-2 tests conducted and stringency index. The section comparing the models now discusses total effect estimated by both models. This new addition widened the estimates, and these were updated in the abstract and the discussion. However, these still might be conservative and so this has been added to the limitations.

(Discussion, line 350)

Modelling both pathogens brings many challenges; … Considering all of these challenges, our estimates might be conservative.

• Comment 2: Similarly, for Campylobacter the decrease is from ~1,000 to 250 which corresponds to a 75% reduction. But again, the estimates are lower.

Response: Similarly, to the response above, there is a lot of uncertainty, e.g. seasonality can differ by region (see Louis et al., 2005). In our paper, the estimate is provided as a mean value over the 32 weeks. The estimated impact at the beginning was between 38% – 55% but then it decreased exponentially. The actual difference between the lines is ~34% during the pandemic and the MAE in the period before the pandemic was ~76 reports. Clarifications were added to the manuscript (see highlights below).

Changes in the manuscript (Materials and methods, lines 165-170):

… For example, level shift is the same on the first as well as the fifth or tenth week, while for transient shift, the highest impact is assumed at the beginning, but decreases exponentially with time, e.g. (δ = 0.85) in the second week but (δ = 0.52) in the fifth week (see Fig. 1). The estimated coefficient of the indicator is then a relative change in the weekly laboratory reports of a given pathogen, considering the other variables and effects in the model such as air temperature, seasonality and autoregression.

• Comment 3: I agree with the authors assessment that delta=1 for norovirus and approximately 0.85 for Campylobacter but they do not provide any justification for the later value. You could argue, for example, that the reduction in Campylobacter halves over 4 weeks and 0.85^4=0.52.

Response: The reduction for Campylobacter indeed halves over 4 weeks. As described in the methods, we compare level and transient shifts. The transient shift represents exponential decay. To clarify this more in the paper, a plot visualising the individual effects was added as well as specific examples in the text.

Changes in the manuscript (Results, lines 224-228):

The decrease in Campylobacter reports was better described by transient shift (δ = .85). The estimated impact was ~46% (CI 38% - 55%) on week 12, ~39% (32% - 47%) on week 13 and so on; the mean weekly reduction across weeks 12 and 43 was ~9% (CI 8% - 11%). This decrease was statistically significant at the 10% but not the 5% level (p = 0.075). 

• Comment 4: It is a bit of shame that delta is not estimated from the data. Fitting a transfer function model would be one way to do this - see for example chapter 13 of “Time Series Analysis” by Box, Jenkins , Reinsel and Ljung (a pdf version of the book is available for free online)

Response: As the paper aims to demonstrate not only quantitative, but also qualitative difference between the impacts of the pandemic on norovirus and Campylobacter, I think the current format of the analysis achieves this better, i.e. it allows for easier communication of the message – Norovirus is more similar to Sars-CoV-2 than Campylobacter and so the reduction in norovirus is related to the actual measures, not only changes to the health seeking behaviour. It is our feeling that a more detailed analysis along the constructive lines suggested is beyond the scope of our paper.

---

## [Decision Letter · Decision Letter 1]

30 Jul 2021

PONE-D-21-11454R1

Differential impact of the COVID-19 pandemic on laboratory reporting of norovirus and Campylobacter in England: a modelling approach

PLOS ONE

Dear Dr. Ondrikova,

Thank you for submitting your manuscript to PLOS ONE. After careful consideration, we feel that it has merit but does not fully meet PLOS ONE’s publication criteria as it currently stands. Therefore, we invite you to submit a revised version of the manuscript that addresses the points raised during the review process.

We look forward to receiving your revised manuscript.

Kind regards,

Shinya Tsuzuki, MD, MSc

Academic Editor

PLOS ONE

Journal Requirements:

Additional Editor Comments:

Both reviewers made positive decisions but one of them raised a minor concern. Please answer the comment before publication.

Reviewers' comments:

Reviewer's Responses to Questions

**Comments to the Author**

1. If the authors have adequately addressed your comments raised in a previous round of review and you feel that this manuscript is now acceptable for publication, you may indicate that here to bypass the “Comments to the Author” section, enter your conflict of interest statement in the “Confidential to Editor” section, and submit your "Accept" recommendation.

Reviewer #1: All comments have been addressed

Reviewer #2: All comments have been addressed

2. Is the manuscript technically sound, and do the data support the conclusions?

Reviewer #1: Partly

Reviewer #2: Yes

3. Has the statistical analysis been performed appropriately and rigorously? 

Reviewer #1: Yes

Reviewer #2: Yes

4. Have the authors made all data underlying the findings in their manuscript fully available?

Reviewer #1: Yes

Reviewer #2: Yes

5. Is the manuscript presented in an intelligible fashion and written in standard English?

Reviewer #1: Yes

Reviewer #2: Yes

6. Review Comments to the Author

Reviewer #1: (No Response)

Reviewer #2: I would like to thank the authors for addressing my comments. In my view the paper is acceptable for publication. I have one question and a suggestion.

Question:

I was unsure how to interpret the norovirus shift parameter estimate in the new Table 3. I imagined exponentiating this parameter would give me the measure of impact but it doesn't (exp(-0.792)=0.45 but the reported impact is 59%). Is this an error or how should this parameter be interpreted?

Suggestion:

Given that seasonality is so important in this analysis it might be worth illustrating it somehow - e.g. by promoting the weekly time series figures from the supplementary information to the main text.

7. PLOS authors have the option to publish the peer review history of their article (what does this mean?). If published, this will include your full peer review and any attached files.

Reviewer #1: No

Reviewer #2: **Yes: **Christian Bottomley

---

## [Author Response · Author response to Decision Letter 1]

10 Aug 2021

We appreciate the extra effort that the reviewers have dedicated to providing additional feedback. We have incorporated changes to the manuscript and one of the figures to reflect the comments provided. 

Here is a point-by-point response to the comments. 

Comments from Reviewer 2

• Question: I was unsure how to interpret the norovirus shift parameter estimate in the new Table 3. I imagined exponentiating this parameter would give me the measure of impact but it doesn't (exp(-0.792)=0.45 but the reported impact is 59%). Is this an error or how should this parameter be interpreted? 

Response: Thank you for spotting this, it needs clarification in the text. Briefly, since three variables represent the impact in the second model, the estimate is a sum of these three variables giving us ~59% decrease. 

Specifically, all coefficients are transformed by exponentiating and subtracting one – (exp(coefficient) – 1) * 100. So the shift is then (exp(-0.792) – 1) * 100 ~ 55%. In the second model, we have two more variables indicating impact, which are added to the shift. As stringency index and the number of conducted Sars-CoV-2 tests vary over time, we need to multiply the coefficients by the actual values of the respective variable. Additionally, we only consider the period between weeks 12 and 43 to match the timeline of the shift variable. For norovirus, both the conducted tests and stringency index give us a decrease of approximately 2%, which is 4% in total and 55% + 4 % gives 59%. The simpler overall model represents the pandemic only with the shift indicator where (exp(-0.880) -1) *100 ~59%.

Changes in the manuscript (Results, lines 271-281):

… A simpler (overall) model might have underestimated the impact as the total decrease in norovirus reports determined by the second (specific trends) model has wider confidence intervals ~59% (CI 47% - 79%). On the other hand, the point estimates of both the Overall Impact and Specific Trends models were ~59%. Similarly, the total mean reduction of Campylobacter reports estimated by the Specific Trends model was between ~11% (CI 8% - 17%), which is higher than the simpler Overall Impact model ~9% (8% - 11%). Note that the impact estimate from the Specific Trends models is a sum of all the pandemic related estimates, i.e. conducted Sars-CoV-2 tests, stringency index and the shift determined by the overall impact model.

• Suggestion: Given that seasonality is so important in this analysis it might be worth illustrating it somehow - e.g. by promoting the weekly time series figures from the supplementary information to the main text.

Response: Thank you for the suggestion. Figure 2 now includes the previous season (2018/19), showing two years in total, and the seasonality is visible there.

---

## [Editor Report · Decision Letter 2]

12 Aug 2021

Differential impact of the COVID-19 pandemic on laboratory reporting of norovirus and Campylobacter in England: a modelling approach

PONE-D-21-11454R2

Dear Dr. Ondrikova,

We’re pleased to inform you that your manuscript has been judged scientifically suitable for publication and will be formally accepted for publication once it meets all outstanding technical requirements.

Kind regards,

Shinya Tsuzuki, MD, MSc

Academic Editor

PLOS ONE
---

## [Editor Report · Acceptance letter]

16 Aug 2021

PONE-D-21-11454R2 

Differential impact of the COVID-19 pandemic on laboratory reporting of norovirus and *Campylobacter* in England: a modelling approach 

Dear Dr. Ondrikova:

I'm pleased to inform you that your manuscript has been deemed suitable for publication in PLOS ONE. Congratulations! Your manuscript is now with our production department. 

Kind regards, 

on behalf of

Dr. Shinya Tsuzuki 

Academic Editor

PLOS ONE